# Quality Control Optimization for Minimizing Security Risks Associated with Mesenchymal Stromal Cell-Based Product Development

**DOI:** 10.3390/ijms241612955

**Published:** 2023-08-19

**Authors:** Carmen Lúcia Kuniyoshi Rebelatto, Lidiane Maria Boldrini-Leite, Debora Regina Daga, Daniela Boscaro Marsaro, Isadora May Vaz, Valderez Ravaglio Jamur, Alessandra Melo de Aguiar, Thalita Bastida Vieira, Bianca Polak Furman, Cecília Oliveira Aguiar, Paulo Roberto Slud Brofman

**Affiliations:** 1Core for Cell Technology, School of Medicine, Pontifícia Universidade Católica do Paraná, Curitiba 80215-901, Brazil; lidiane.leite@pucpr.br (L.M.B.-L.); regina.daga@pucpr.br (D.R.D.); marsaro.daniela@pucpr.br (D.B.M.); isadora.may@pucpr.br (I.M.V.); valderez.ravaglio@pucpr.br (V.R.J.); thalita.vieira@pucpr.edu.br (T.B.V.); bianca.furman@pucpr.edu.br (B.P.F.); cecilia.aguiar@pucpr.edu.br (C.O.A.); paulo.brofman@pucpr.br (P.R.S.B.); 2National Institute of Science and Technology for Regenerative Medicine—INCT-REGENERA, Rio de Janeiro 21941-599, Brazil; 3Laboratory of Basic Biology of Stem Cells, Carlos Chagas Institute—Fiocruz-Paraná, Curitiba 81350-010, Brazil; alessandra.aguiar@fiocruz.br

**Keywords:** good manufacturing practice, advanced therapy medicinal product, umbilical cord

## Abstract

Mesenchymal stromal cells (MSCs) have been considered a therapeutic strategy in regenerative medicine because of their regenerative and immunomodulatory properties. The translation of MSC-based products has some challenges, such as regulatory and scientific issues. Quality control should be standardized and optimized to guarantee the reproducibility, safety, and efficacy of MSC-based products to be administered to patients. The aim of this study was to develop MSC-based products for use in clinical practice. Quality control assays include cell characterization, cell viability, immunogenicity, and cell differentiation; safety tests such as procoagulant tissue factor (TF), microbiological, mycoplasma, endotoxin, genomic stability, and tumorigenicity tests; and potency tests. The results confirm that the cells express MSC markers; an average cell viability of 96.9%; a low expression of HLA-DR and costimulatory molecules; differentiation potential; a high expression of TF/CD142; an absence of pathogenic microorganisms; negative endotoxins; an absence of chromosomal abnormalities; an absence of genotoxicity and tumorigenicity; and T-lymphocyte proliferation inhibition potential. This study shows the relevance of standardizing the manufacturing process and quality controls to reduce variability due to the heterogeneity between donors. The results might also be useful for the implementation and optimization of new analytical techniques and automated methods to improve safety, which are the major concerns related to MSC-based therapy.

## 1. Introduction

Mesenchymal stromal cells (MSCs) have been considered a new therapeutic strategy in regenerative medicine [1]. These cells have emerged as a perspective for the development of Advanced Therapy Medicinal Products (ATMPs) mainly because of their regenerative and immunomodulatory properties [2]. According to Brazilian National Health Surveillance Agency (ANVISA) regulations, Advanced Therapy Medicinal Products are products subjected to substantial manipulation so that their biological characteristics, physiological functions, or structural properties, relevant for the intended clinical use, have been modified or if they are not intended to be used for the same essential function(s) in the recipient and the donor [3].

Mesenchymal stromal cells have great potential in regenerative medicine because they simultaneously activate multiple mechanisms, such as paracrine, trophic, immunomodulatory, and differentiation mechanisms, affecting all stages of the regeneration of damaged tissues [4]. Furthermore, MSCs are easily isolated from different tissues, have a high migratory capacity, have high expansion rates, and have the ability to avoid allogeneic responses after transplantation [5,6].

Currently, few MSC-based products have acquired marketing authorization [7] because they represent considerable regulatory and scientific challenges in translation [8,9,10,11]. Although health agencies have published specific guidelines to define the criteria for cell-based products, which must be implemented in the manufacturing of MSC-based products, there are multiple obstacles for the transfer from research use to the pharmaceutical grade of MSCs. Regarding scientific issues, according to the International Society for Cell & Gene Therapy (ISCT), it is critical to have an extensive phenotypic characterization, trophic factor expression, potential mechanisms of action, and quality controls for any cells used in regenerative medicine [12]. Some critical points, such as cell isolation, in vitro expansion, validation, and quality control, should be standardized and optimized to guarantee reproducibility and MSC quality [3]. Quality control is extremely important for an ATMP because it enhances its safety and efficacy for administration to patients [13]. Quality control should be implemented throughout the manufacturing process by evaluating the biological samples, intermediate products, and final product before release to ensure that the MSC-based final product consistently meets its specifications across batches [14].

One of the major challenges in the development of MSC-based products is the large variability depending on the sources currently used for their isolation [15]. Different sources exhibit different marker profiles, gene expressions, differentiation potentials, and immunomodulatory and paracrine properties [16,17]. Other factors are the donor, which could be related to age and health, among other factors, and the manufacturing process, which includes the composition of the culture medium and supplements, the substrate properties, and the oxygen concentration and would most likely affect the features of MSCs through different mechanisms [18,19,20]. All this variability is an issue in the development of an ATMP, and therefore, manufacturing standardization and product characterization are extremely important to limit the sources of variability.

Umbilical cord-mesenchymal stromal cells (UC-MSCs) are considered promising candidates for developing allogeneic MSC-derived ATMPs because they are easily accessible sources of the starting material, with noninvasive collection, few ethical concerns, as they are usually discarded at birth, and higher proliferation capacities [21,22,23]. All these features offer the possibility of generating master cell and work cell stocks to facilitate manufacturing and to produce several batches from one donor [14]. Furthermore, UC-MSCs are weakly immunogenic, thus completing the MSC safety profile that was demonstrated in several clinical trials [23,24]. These advantages show the wide range of potential applications for UC-MSCs [25,26,27]. However, UC-MSCs are heterogeneous cells, and the phenotype may be influenced by the manufacturing process, which could be the reason for the varied therapeutic outcomes from different laboratories [4,28,29].

Efforts to improve the performance of next-generation MSC-based products are necessary to achieve optimal safety and efficacy. Umbilical cord-MSCs are a promising treatment option for a variety of clinical conditions, and some potential risks can be decreased if robust and strict quality controls are performed, ensuring safety in clinical trials. In this context, the aim of this study is to develop a UC-MSC-based product according to Good Manufacturing Practice (GMP) for use in clinical practice.

In the present work, the manufacture of a UC-MSC-based product for use in the treatment of inflammatory disease was demonstrated. It was possible to demonstrate a reproducible manufacturing process and the optimization of quality controls by implementing new methodologies to ensure that the final product consistently meets its specifications across batches and improves the safety of UC-MSCs.

## 2. Results

The results from five fresh UC-MSC samples after thawing and cell expansion are shown. In this study, UCs were processed no longer than two hours after collection, and the success rate for isolating UC-MSCs was 100%. During the culture time, the cells adhered to plastic and exhibited a fibroblastoid morphology.

The UC-MSC surface antigen profile was evaluated by flow cytometry. All samples displayed similar immunophenotypes for the markers analyzed. The cells express the markers CD90 (99.82% ± 0.192), CD105 (98.42% ± 0.654), CD73 (96.78% ± 0.76), and CD29 (99.12% ± 0.409) and lack the expression of CD 14 (1.732% ± 0.15), CD19 (0.934% ± 0.331), CD34 (0.308% ± 0.147), CD45 (0.262% ± 0.188), and HLA-DR (1.286% ± 0.588) (Figure 1A). Considering immunogenicity evaluation, UC-MSCs expressed HLA class I (93.08% ± 1.878) and a low expression of HLA-DR (1.286% ± 0.588) and the costimulatory molecules CD40 (0.478% ± 0.205), CD80 (0.572% ± 0.289), and CD86 (0.752% ± 0.237) (Figure 1B). The average expression of TF was 82.96% (range 76.7–92.2%), and great variability was observed between different donors (Figure 1C). The cell viability was 96.9% ± 1.715, and annexin V presented reduced values of 0.69% ± 0.383 (Figure 1D).

Differentiation into adipocytes, osteoblasts, and chondrocytes was qualitatively assessed on the basis of cell morphology and cytochemistry. All UC-MSC samples showed the potential to differentiate into adipocytes, characterized by the presence of lipidic vacuoles stained with Oil Red O; osteoblasts, characterized by the presence of calcium deposits stained with Alizarin Red S; and chondroblasts, characterized by the presence of lacunae around young chondrocytes and proteoglycan in the matrix (Figure 2A–F). Osteogenic quantification showed that the induced differentiation samples had higher absorbance than their controls (osteogenic differentiation vs. control: 3.96 ± 0.06987 vs. 0.5163 ± 0.26, *p* < 0.0001) (Figure 2G). Adipogenic quantification was performed; therefore, no differences were observed in relation to the control due to the low potential of UC-MSCs for adipogenic differentiation.

The ATMP was negative for microbiological tests and showed an absence of mycoplasma contamination and endotoxins ≤ 0.5 EU/mL.

To evaluate genetic stability, a G-banding karyotype test and CBMN assay were performed. Metaphases were obtained in all cases and were identified as normal diploid karyotypes. A G-banding karyotype test showed an absence of clonal chromosomal aberrations. In the CBMN assay, the NDI values of the two negative control samples were 1.23 and 1.33. In one sample, the presence of one nucleoplasmic bridge in 1000 cells was observed. The positive control NDI (HeLa cells) was 1.5, with cells showing the presence of several micronuclei, nucleoplasmic bridges, and nuclear buds. The NDI values for the five UC-MSC samples were 1.3, 1.17, 1.23, 1.29, and 1.37. In one sample, the presence of two cells with nucleoplasmic bridges in 1000 cells was shown. These results show that the cells divided, allowing for the protocol efficacy (Figure 3).

There were no significant differences between the UC-MSC and negative control groups (micronucleus: 0 ± 0 vs. 0 ± 0, *p* > 0.999; nucleoplasmic bridges: 0.4 ± 0.894 vs. 0.5 ± 0.577, *p* = 0.841; nuclear buds: 0 ± 0 vs. 0 ± 0, *p* > 0.999). The results show that the culture conditions and cryopreservation were not genotoxic for UC-MSCs.

The soft agar colony formation assay was performed to assess in vitro cellular anchorage-independent growth, i.e., the ability to proliferate unattached to extracellular matrix and neighboring cells. This is a method for detecting the malignant transformation of cells and tumorigenicity. The two protocols showed similar results. In the standard soft agar colony formation assay, after 21 days of culture, UC-MSCs did not generate any colonies, while HeLa cells, used as a positive control, formed a large number of colonies (Figure 4).

In the fluorescence cell growth assessment protocol, the RFU of UC-MSC samples at different cell concentrations was significantly smaller than that of HeLa cells (1000 cell/well—MSC vs. HeLa: 4205 ± 1608 vs. 13,990 ± 2893, *p* < 0.0001; 5000 cell/well—MSC vs. HeLa: 11,404 ± 5852 vs. 51,721 ± 19,208, *p* < 0.0001; 10,000 cell/well—MSC vs. HeLa: 16,315 ± 9128 vs. 98,701 ± 8338, *p* < 0.0001) (Figure 5).

The inhibition of T-lymphocyte proliferation is a potency test for confirming the immunomodulatory properties of UC-MSCs. A potency assay using the mixed lymphocyte reaction (MLR) using different ratios of the PBMC:UC-MSC coculture was performed to assess whether the inhibitory effect of UC-MSCs on T-lymphocyte proliferation is dose-dependent. T-lymphocyte growth was inversely proportional to UC-MSC quantity. The T-lymphocyte proliferation was significantly different compared to that of control cells (T lymphocyte + polyclonal stimulus) at ratios of 1:2 vs. the control (40.63 ± 8.566 vs. 78.40 ± 2.427, *p* < 0.0001), 1:5 vs. the control (32.79 ± 10.86 vs. 78.40 ± 2.427, *p* < 0.0001), and 1:10 vs. the control (21.87 ± 13 vs. 78.40 ± 2.427, *p* < 0.0001). There was a difference between the concentrations at ratios of 1:2 vs. 1:10 (40.63 ± 8.566 vs. 21.87 ± 13, *p* = 0.0001) and 1:5 vs. 1:10 (32.79 ± 10.86 vs. 21.87 ± 13, *p* = 0.0456). The inhibition of T lymphocytes by UC-MSCs showed the following results according to the concentrations: 50.608 ± 8.808 (1:2); 58.18 ± 13.52 (1:5), and 72.11 ± 11.72 (1:10). Therefore, the inhibition potential increased in the presence of UC-MSCs in a dose-dependent manner (Figure 6).

The results displayed in this study confirm that all UC-MSC samples are in accordance with the acceptance criteria for the clinical use of ATMPs established at CTC/PUCPR. These criteria included cell viability ≥70%; the positive expression markers (≥95%) CD73, CD90, CD105, and CD29; the negative expression (≤2%) of CD45, CD34, CD14, CD19, and HLA-DR; a low expression of HLA-DR and the costimulatory molecules CD40, CD80, and CD86; the absence of contamination with pathogenic microorganisms (bacteria, mycoplasma, and fungi); endotoxins ≤ 0.5 EU/kg; the absence of clonal abnormalities; and the inhibition of T-lymphocyte proliferation ≥ 50%.

The evaluation of TF/CD142, genotoxicity, and tumorigenicity are not mandatory quality controls according to Brazilian regulations [3]. There are no reference values, and therefore, they are not included in the quality control criteria for releasing ATMP for clinical use.

## 3. Discussion

In recent years, MSCs have been used as a new therapeutic strategy to treat several diseases, although the development of MSC-based products is very complex [30]. One of the challenges is the standardization of the manufacturing process, where the definition of which quality controls to perform and when they must be performed is an important issue, since these factors will affect the safety, efficacy, and product-related costs. This study showed a reproducible manufacturing process of an investigational UC-MSC-based ATMP authorized for clinical use and the optimization of quality controls by implementing new methodologies that minimize security risks associated with UC-MSC-based product development. There is a need to define more stringent, specific, and harmonized requirements to characterize the quality of MSCs and enhance the analysis of their safety and efficacy in final products to be administered to patients [13]. Some quality control tests are already well established in the regulation regarding the therapeutic use and clinical research of MSCs. However, some evaluations have no detailed published guidelines, such as genotoxicity and tumorigenicity. Therefore, this work also suggests additional tests for MSC-based products for monitoring chromosomal instability and tumorigenicity during cell expansion.

In this research, UC-MSCs cryopreserved in a biobank were used for clinical trials. According to the results presented, UC-MSCs maintained the characteristics of MSCs, showing that cryopreservation does not change their biological properties, which is an optimal solution in terms of future clinical use. This is in line with other research [31,32].

In this work, an enzymatic digestion protocol was performed. Mori et al. [33] showed that the enzymatic digestion protocol increases the cell recovery rate. Furthermore, some reports revealed that the explant method allows for the selection of a cell fraction with higher proliferative potential [34,35], but a remarkable variation in cell phenotype expression was observed compared to enzymatic digestion [36,37]. In the five analyzed UC-MSC samples, the isolation efficacy of the primary cell culture was 100%. These cells showed a spindle fibroblast-like cell morphology and the capacity to adhere to culture flasks without notable morphologic changes during expansion. They also exhibited high proliferation abilities, and according to Voisin et al. [23], this is essential for MSCs to maintain stemness characteristics during in vitro culture. In this research, UC-MSCs were cultured until passage five. It is recommended to perform a reduced number of population doubling conditions to avoid the occurrence of cytogenetic anomalies [38].

The quality controls to be performed on an MSC-based ATMP include at least tests of identity, purity, safety, and potency. As part of cell characterization, cell expression profiling was performed by flow cytometry. This technique helps to define and validate either the homogeneity or heterogeneity (i.e., potential contamination) of MSCs in a tissue-specific way [15]. The UC-MSC marker expression is in conformity with ISCT criteria and other studies characterizing this type of source [4,39]. The low immunogenicity of MSCs, an important feature of these cells that allows for their use in an allogeneic context, was evaluated by surface marker expression. The results showed that UC-MSCs remained hypoimmunogenic because they showed a low expression of major histocompatibility complex class II (MHC-II) and the costimulatory molecules CD80, CD86, and CD40, which are important for T-cell activation. The low immunogenicity of UC-MSCs was also demonstrated by other researchers [40,41]. In this study, the expression of TF (CD142), a potent activator of coagulation, was assessed to verify the potential risk of thromboembolic complications upon UC-MSC intravenous infusion. Furthermore, the procoagulant activity depends on culture handling conditions and donor variation [42]. UC-MSC samples showed a high expression of TF markers. Variability between the different samples was also observed, showing that all MSC populations were not equivalent. Christy et al. [42] reported great differences in the procoagulant function between different MSC populations and a strong correlation between the percentage of cells expressing surface TF and functional procoagulant activity. Therefore, to ensure the safety of the product, care in selecting cells and other solutions, such as the use of anticoagulants, must be taken for clinical use. The results displayed in this work reinforce the integration of TF/CD142 expression as a criterion for intravascular MSC products [43].

Cell differentiation, which must also be evaluated for MSC characterization, is a highly regulated process that depends on various extracellular and intracellular factors of its modulation [15]. In this study, it was demonstrated that in vitro UC-MSCs have the potential to differentiate into mesoderm-type lineages, including adipogenic, osteogenic, and chondrogenic lineages, as observed by other studies [4,23,44,45,46,47,48,49,50,51]. UC-MSCs presented a low potential for adipogenic differentiation. Researchers have shown that some factors, such as donor, tissue source, and epigenetic regulation factors, also determine the differentiation potential of MSCs [17,20]. Other studies also reported that Wharton Jelly MSCs (WJ-MSCs) weakly differentiated into adipocytes [52,53]. In this research, variability in the adipogenic potential between samples after differentiation was also observed. Arutyunyan et al. [4] observed that the plasticity of UC-MSCs may depend on the conditions of pregnancy, such as impaired metabolism of the maternal organism during pregnancy, which has a significant impact on the biological properties of neonatal MSCs [54,55]. The plasticity of UC-MSCs, depending on the cord donor, could explain this variability.

It is important to note that cells cultured in vitro are subjected to selective pressures, which may alter their immunomodulatory properties [36,56,57,58]. Furthermore, different potency tests may be required depending on the intended disease target. The MSC immunomodulatory potential is the most commonly used potency test due to the use of these cells for the treatment of inflammatory diseases [59]. In this study, mixed lymphocyte reaction assays were performed to evaluate the suppression of T-cell proliferation. It was shown that UC-MSCs retained their immunomodulatory potential with the protocol used in this study. This is consistent with other studies [60,61]. Mattar and Bieback [16] described that WJ-MSCs have stronger immune capacities than bone marrow MSCs and are an important source of MSCs for the treatment of immune-mediated inflammatory diseases.

ATMP development is not based on fixed preestablished criteria, but it should be adaptable case-by-case to address the complexity and heterogeneity of cell therapies, which is why the reliable quality control of MSCs is elusive [62]. What is strongly recommended is a risk-based approach covering the entire development of ATMPs [63]. Mesenchymal stromal cells cultivated in vitro can lead to an accumulation of genetic and epigenetic alterations, featuring genetic instability that may explain their tumorigenic potential [64,65]. Evaluating chromosomal instability during a cell culture is essential, and the methods of choice are still challenging due to the execution time, the release of results, and the costs. The evaluation is usually carried out by cytogenetics. However, additional assays, such as the CBMN and tumorigenicity assays, for assessing the consequences of this instability, make it possible to evaluate the process more completely, since cell changes in cells can occur through different processes or at different stages. Therefore, in this research, additional assays that increase product safety and enable high-throughput screening and a low cost were performed to guarantee the quality control of MSCs.

The evaluation of genotoxicity and tumorigenicity are not mandatory quality controls according to Brazilian regulations [3]; therefore, it is recommended that the genetic stability and tumorigenicity potential of the cells be demonstrated. Conventional karyotyping and other techniques should be combined to evaluate genomic integrity [66]. In this study, karyotyping analysis demonstrated the maintenance of stability up to P5 of the cultured UC-MSCs and the absence of clonal abnormalities. These results are in line with other studies that observed a normal diploid karyotype in UC-MSCs up to P6 and did not find any instability until that passage [67,68].

The assessment of genotoxicity allows for verifying if any of the reagents or materials that came into contact with the cells during the ATMP manufacturing process could have a genotoxic effect, namely, damaging genetic information in cells and consequently affecting the product quality [69]. The CBMN assay can detect abnormalities such as bridges, buds, and micronuclei, which may be related to a risk of developing cancers, since they are indicators of instability genetics [70,71]. In this study, there were no significant differences between UC-MSCs and normal controls, suggesting that culture conditions maintain the genetic stability of cells. These results are consistent with those of Sharma et al. [69], who observed the presence of higher quantities of micronuclei and buds in HeLa cells than in UC-MSCs. They also described the genetic stability of UC-MSCs until higher passages, using micronuclei for genotoxic issues. In addition, UC-MSCs present a lower number of micronuclei and nuclear blebs than human placenta-derived MSCs, even when isolated from the same patient, which means that UC-MSCs are genetically stable [72,73].

As a suggestion, based on the results presented in this study, the CBMN assay could be used as an additional quality control test to assess genomic integrity, increasing the safety of the final product. To date, there are no reference values for this assay; therefore, in this research, no significant difference compared to the normal control was considered to indicate the absence of genotoxicity.

Genomic instability could lead to the development of tumors, ectopic tissue formation, and/or malignant transformation [13]. Therefore, the evaluation of the tumorigenic potential of expanded MSCs is recommended. Tumorigenicity can be evaluated by in vivo assays, using immunocompromised animals, and by in vitro assays. In vivo assays are very time-consuming, and in vitro assays would be a good option for ATMP development. Furthermore, it was demonstrated that there is a correlation between in vivo and in vitro findings [73]. A soft agar colony formation assay is indicated for detecting the malignant transformation of cells and allowing for the assessment of the presence and quantification of colonies [74]. In this study, two methods were used to assess tumorigenicity, and both showed similar results, with an absence of colony formation in UC-MSC samples, suggesting that UC-MSCs are chromosomally stable under culture conditions. These results are in accordance with other studies that demonstrated that human-expanded UC-MSCs do not possess anchorage-independent proliferation potential [75,76]. These cells are nontumorigenic and have the potential for safe cell-based therapies. Both methods used are quantitative, although in the standard soft agar assay, the colonies are counted manually, which is highly subjective. Another difficulty is that, most of the time, the colonies are spread, making them difficult to count. For this reason, the quantitative cell transformation assay, through fluorescence microplate readers, showed some advantages: it is more sensitive, allows for a high-throughput screening of formed colonies, is faster, and allows for the analysis of larger sample sizes simultaneously. This assay showed more reliable results since it would meet the quality assessment criteria of ATMPs and will be useful for quality control in the manufacturing process. There are no established reference values that exclude the potential for cell tumorigenicity; therefore, in this study, the absence of a significant difference in RFU between the positive control and UC-MSCs was considered a criterion for the absence of transformation potential.

Briefly, it was shown that the ATMP, a UC-MSC-based product, was safe considering its sterility, which includes microbiology, mycoplasma, and endotoxin testing; the maintenance of genomic stability during in vitro expansion by karyotype analysis and CBMN assays; and the absence of tumorigenicity. The bioactivity assessed by immunomodulation showed that these cells maintain their functionality and might be a useful product for clinical use.

There were some limitations regarding the manufacturing process. Some nonpharmaceutical-grade raw materials, such as FBS, trypsin, and collagenase, were used, and consequently, they were not free of xenogenic compounds, which is recommended for the GMP process [14]. Although the use of FBS-supplemented media has been the gold standard for cell cultures and is allowed for clinical use, the presence of animal-derived proteins could stimulate an immune response [77]. Future studies of MSC-based product manufacturing process standardization using clinical grade reagents should be performed. They will provide safer MSC products for cell therapy.

This manuscript describes the cell therapy products pipeline. It makes essential contributions, showing that the protocol used for the quality control of ATMP in different steps of the manufacturing process allows for reproducibility and meets all the requirements of the safety criteria established by Brazilian and international regulations.

There are few scientific reports on the application of regulatory criteria in ATMP quality control. Therefore, this research addresses the translational gap between regulatory frameworks related to ATMP and laboratory practice, bringing important contributions to the clinical development of ATMP. Furthermore, this study suggested additional tests for evaluating the procoagulant tissue factor and genetic instability to optimize quality controls and minimize security risks associated with MSC-based product development. More studies are needed to define the reference values of these methods so that they can be routinely used as quality control assays for ATMP.

## 4. Materials and Methods

### 4.1. Study Design

According to ANVISA regulations, all data about the manufacturing process must be submitted to regulatory authorities, which review the information and approve the protocol, if appropriate [3]. The protocol and quality controls described in this work are being used in a study approved by ANVISA for the treatment of pneumonia in patients with SARS-CoV-2 (N° 25351.076011/2021-44).

The UC was used as a source of MSCs. Five samples were obtained from full-term neonates and processed at Cell Core Technology (CCT) from the Pontificia Universidade Catolica do Parana (PUCPR). The mother of the donor signed an informed consent form approved by the institutional review board (Ethics Committee—CAAE: 31935420.7.1001.0020). Healthy umbilical cord donors underwent serological testing for infectious diseases and social, laboratory, and clinical screening. The study was conducted following the GMP guidelines for ATMP by highly trained staff and in appropriate and monitored facilities. The UC-MSC-based product manufacturing process encompasses different stages (Figure 7).

In accordance with Brazilian regulations, quality controls were performed in different phases of UC-MSC manufacturing, such as before cryopreservation for storage in a mesenchymal cell biobank (master cell stock) and after thawing the cells, in the final product (Table 1). In this study, the cells were thawed and expanded for one more passage, and quality controls were performed. Cryoprotective agents show cell toxicity; therefore, quality control evaluation after thawing is essential during the manufacturing process to demonstrate that cells maintain their characteristics and functionality potential after cryopreservation.

All quality controls were performed between passages (P) three and five (P3–P5). At CTC/PUCPR, passage five is considered the last one for the clinical use of UC-MSCs since cells showed high proliferative capacities without chromosomal abnormalities at this stage.

### 4.2. Advanced Therapy Medicinal Product

Umbilical cords were obtained from full-term newborns by cesarean sections and were aseptically stored in sterile Iscove’s modified Dulbecco’s medium (IMDM) (Gibco BRL, Grand Island, NY, USA) supplemented with 100 U/mL penicillin and 100 µg/mL streptomycin (Gibco BRL, Grand Island, NY, USA). The umbilical cord was washed three times with phosphate buffered saline (PBS) (Gibco BRL, Grand Island, NY, USA) and antibiotics, sectioned into small fragments, transferred to conical tubes with PBS, and centrifuged at 400× *g* for 10 min. After removing the supernatant fraction, the tissue was treated with 0.1% collagenase type II (Sigma, St. Louis, MO, USA) under constant homogenization at 37 °C for 16 h, washed with PBS, and further digested with 0.25% trypsin-EDTA (Gibco, Grand Island, NY, USA) at 37 °C for 15 min. Fetal bovine serum (FBS) (HyClone™, South Logan, UT, USA) was added to the MSCs to neutralize the trypsin [79], and the sample was centrifuged at 400× *g* for 10 min.

The cells were resuspended and plated in 75 cm^2^ culture flasks (Greiner Bio-One, Kremsmünster, Austria) with IMDM supplemented with 20% FBS and antibiotics and incubated at 37 °C and 5% CO_2_. At 72 h, nonadherent cells were removed and washed with PBS, and the culture medium was replaced with fresh medium once a week. When the cell culture reached 80% confluence, the cells were detached by enzymatic dissociation treatment. The cells were washed with PBS, 0.25% trypsin-EDTA was added, and the cells were placed in an incubator at 37 °C for four minutes. After this period, SBF was added to neutralize the trypsin. The cells were collected and transferred to conical tubes and centrifuged at 400× *g* for 10 min. The supernatant was discarded, and the cells were resuspended and replated into 150 cm^2^ culture flasks.

For cryopreservation of the master cell stock, a rate-controlled freezer (Thermo Scientific, Waltham, MA, USA) was used at a final concentration of 10% dimethyl sulfoxide (Origen, Austin, TX, USA) and 90% FBS. The cells were transferred to a container with liquid nitrogen until use for the assays. Between P2 and P3, quality controls were performed. The cells were thawed in a water bath at 37 °C, transferred to conical tubes with IMDM and 10% SBF, and centrifuged at 400× *g* for 10 min. The supernatant was discarded, and the cells were resuspended and replated into 150 cm^2^ culture flasks. Confluent UC-MSCs were detached by enzymatic dissociation treatment, washed twice with Ringer’s lactate solution, and resuspended in a vehicle solution composed of Ringer’s lactate solution (Halexistar, Goiânia, Brazil) and 5% human serum albumin (Blau Farmacêutica, São Paulo, Brazil), and the samples were collected for quality control.

### 4.3. Quality Control

Quality controls were performed to validate the manufacturing process. For cell characterization, recommendations by the International Society for Cellular Therapy (ISCT) [39] were followed, and the quality controls respected the established rules in the Brazilian regulation for ATMPs [3].

The quality controls include cell characterization, performed by the evaluation of cell surface marker expression, cell viability, immunogenicity, and cell differentiation; safety tests such as procoagulant tissue factor (TF), microbiological, mycoplasma detection, endotoxin, genomic stability, and tumorigenicity tests; and potency tests such as immunomodulation.

#### 4.3.1. UC-MSC Surface Marker Expression, Cell Viability, and Immunogenicity

Immunophenotypic analysis was performed by flow cytometry. A total of 1 × 10^6^ MSCs were distributed per tube. The cells were centrifuged with PBS and incubated with conjugated monoclonal antibodies against the following antigens, all from BD Pharmingen (San Diego, CA, USA): CD29 (allophycocyanin (APC) conjugated); CD73 (phycoerythrin (PE) conjugated); CD90 (PE conjugated); CD105 (APC conjugated); CD14 (fluorescein isothiocyanate (FITC) conjugated); CD19 (FITC conjugated); CD34 (APC conjugated); CD45 (FITC conjugated); and HLA-DR (PE-Cyanine 5 (PE-Cy5) conjugated). Isotype-identical antibodies served as controls for the reactions. All incubations were performed in the dark, at room temperature, for 30 min. After incubation, the cells were centrifuged with PBS and fixed with 1% paraformaldehyde. A total of 100,000 events per sample were acquired using an FACSCalibur flow cytometer (BD Biosciences, San Jose, CA, USA) and analyzed with FlowJo software v8.0.2 (FlowJo, Ashland, OR, USA). The analysis protocol included the removal of threshold debris, and UC-MSCs were initially identified based on forward scatter (FSC) and side scatter (SSC) and the positive markers (> 95% expression) and negative markers (<2% expression), as described [39]. The histogram analysis was based on the isotype control expression for each fluorochrome.

The viability and apoptosis of the cells were determined using a method based on 7-aminoactinomycin D (7-AAD) and Annexin V staining. To evaluate immunogenicity, the cells were incubated with the following monoclonal antibodies: HLA-ABC (FITC conjugated), CD40 (FITC conjugated), CD80 (PE-Cy5 conjugated), and CD86 (APC conjugated). A monoclonal anti-CD142 antibody (PE conjugated) was used to evaluate the procoagulant tissue factor (TF).

#### 4.3.2. Cell Differentiation

UC-MSCs were assessed for their potential to differentiate into adipocytes, osteoblasts, and chondrocytes [80]. For adipogenic and osteogenic differentiation, approximately 16 × 10^3^ cells were seeded on glass coverslips (Sarstedt, Newton, NC, USA) in 24-well plates (TPP, Trasadingen, Switzerland) and transferred to an incubator at 37 °C and 5% CO_2_ until the cell culture reached 80% confluence. Then, the cells were treated with the corresponding commercial differentiation media (Lonza, Basel, Switzerland) three times a week for 21 days. Control cells were kept in IMDM with 15% FCS over the same period. For adipogenic differentiation, induced monolayers were fixed with Bouin’s fixative (Biotec, Labmaster, Paraná, Brazil) for 10 min at room temperature and washed twice with 70% ethanol and twice with Milli-Q water. A solution of 0.5% Oil Red O (Sigma-Aldrich) for 1 h was used to visualize lipid-rich vacuoles. Hematoxylin (Biotec, Paraná, Brazil) was used for nuclear staining. For osteogenic differentiation, the cells were fixed with paraformaldehyde (PFA) for 20 min at room temperature and washed with Milli-Q water. The cells were observed by Alizarin Red S staining at pH 4.2 (Fluka Chemie, Buchs, UK) for 5 min to evaluate calcium accumulation.

A micromass culture was performed to promote chondrogenic differentiation. Briefly, 1 × 10^6^ cells in 1 mL of the medium were centrifuged at 400× *g* for 10 min to form a pellet. Without disturbing the pellet, commercial differentiation media were added (Lonza, Walkersville, MD, USA). The medium was changed three times a week for 21 days. The control cells were cultured at the same time as the induced cells with IMDM and 10% SBF. On day 21, cell aggregates were fixed in 10% formaldehyde for 1 h at room temperature, dehydrated in serial ethanol dilutions, and embedded in paraffin blocks. Paraffin sections were stained for histologic analysis with toluidine blue solution (Sigma-Aldrich, St. Louis, MO, USA) to demonstrate the presence of proteoglycans in the extracellular matrix and the gaps around young chondrocytes. The slides were evaluated under a microscope.

For the quantification of adipogenic and osteogenic differentiation, the absorbance was measured by spectrophotometry [81]. Cells induced to undergo adipogenic differentiation and control cells were washed three times with PBS and fixed with 4% PFA (Sigma-Aldrich, St. Louis, MO, USA) for 15 min. Ethanol (70%) was added for cell permeabilization for 30 s, and Oil Red O dye was added for 10 min at room temperature. The samples were washed with 60% isopropyl alcohol and Milli-Q water. After that, 100% ethanol was added for 5 min. A total of 300 µL of the supernatant was distributed in triplicate in a 96-well plate. As a white standard, 100% ethanol solution was used. Quantification was performed using a microplate reader at an absorbance of 550 nm (VersaMax Microplate Reader™, Molecular Device™, Silicon Valley, CA, USA).

Cells induced to osteogenic differentiation and control cells were washed with PBS and Milli-Q water and fixed with 100% ethanol (Merck, Darmstadt, Germany) for 15 min. Alizarin Red S dye (pH 4.23) was added for 40 min at room temperature and washed with Milli-Q water and PBS, and a solution of 10% acetic acid (Merck, Darmstadt, Germany) and 20% methanol (Merck, Darmstadt, Germany) was added to each well. The plate was placed in an orbital shaker at room temperature for 15 min. A total of 300 µL of the supernatant was distributed in triplicate in a 96-well plate. The absorbance was measured using a microplate reader at 450 nm (VersaMax Microplate Reader™—Molecular Devices, Sunnyvale, CA, USA). As a white standard, a solution of 10% acetic acid and 20% methanol was used.

#### 4.3.3. Microbiological Tests

The sterility of the cell cultures and ATMPs was evaluated by tests for the detection of bacteria, fungi, and mycoplasma. Microbiological control was performed for the detection of possible contamination by bacteria and/or fungi through an automated method in Bact/Alert™ 3D equipment (BioMerieux, Durham, NC, USA). The ATMP was transferred to a Bact/Alert ™ PF Plus culture bottle. The bottle was incubated in Bact/Alert ™ at 37 °C for a period of 14 days.

Mycoplasma detection was performed by two methods: PCR-based and bioluminescent assays. The VenorGeM™ Mycoplasma Detection kit (Sigma-Aldrich) was used for the PCR-based assay. Briefly, 100 µL of the cell culture supernatant was transferred to a microcentrifuge tube, heated at 95 °C for 5 min, and centrifuged for 5 sec to remove cell debris. One tube was prepared for the negative control, one was prepared for the positive control, and one was prepared for each sample. PCR was carried out using 2 μL of the sample supernatant, 2.5 μL of the primer mix, 2.5 μL of 10× reaction buffer supplemented with MgCl_2_ (3.0 mM), 2.5 μL of the internal control DNA (plasmid DNA including mycoplasma-specific primer sequences and an internal sequence of the HTLV-I tax gene with a size of ~191 bp), 15.1 μL of DNA-free water, and 0.4 μL of Jumpstart Taq polymerase. As a negative control, DNA-free water was used instead of the sample supernatant. As a positive control, DNA fragments of the M. orale genome (positive control DNA; yields 270 bp band) supplied by VenorGeM^®^ were used. The amplification of the target sequence followed the following conditions: denaturation at 94 °C for 2 min, followed by 39 cycles of denaturation at 94 °C for 30 s, annealing at 55 °C for 1 min, and extension at 72 °C for 30 s (Gene Amp PCR System 9700, Applied Biosystems, Waltham, MA, USA). The amplified sequence was analyzed by electrophoresis, conducted for 20 min at 100 V, on a 1.5% agarose gel stained with SYBR Safe (Invitrogen, Waltham, MA, USA), using 5 µL of the amplified product with 1 µL of Gel Loading Solution (Sigma-Aldrich). The products were visualized in an LED transilluminator (LED 001, Kasvi, PR, Brazil).

The bioluminescent assay was a complementary test for PCR because it was faster, and the results were available when ATMP was released for clinical use. For this assay, the MycoAlert™ PLUS Kit Mycoplasma Detection (Lonza, Rockland, WA, USA) was used according to the manufacturer’s instructions. The culture medium was collected and centrifuged for 5 min at 200× *g*, and the supernatant was analyzed on the equipment (Lucetta, Lonza, Rockland, WA, USA). For each assay, negative and positive controls and the sample were analyzed. The reagent and the substrate were added to each tube. Two readings were performed, before and after adding the substrate. The relationship between the second and the first reading indicates the presence or absence of viable mycoplasmas.

#### 4.3.4. Purity Test

Endotoxins are lipopolysaccharides from Gram-negative bacteria, and their presence in any product implies contamination by pyrogenic components. Endotoxin detection was performed by the Endosafe^®^ Portable Test System (PTS™) (Charles River, Charleston, SC, USA) according to the manufacturer’s instructions. The culture medium was collected and centrifuged for 10 min at 400× *g*, and the supernatant was analyzed on the equipment. For clinical application, the analysis should be <5.00 EU/kg.

#### 4.3.5. Genomic Stability

Cytogenetic analyses for evaluating genomic stability were performed by the G-Banding Technique [82]. Forty-eight hours after cell expansion, cell confluence was evaluated, and the ideal condition for cytogenetic harvest was when the culture reached 60% to 80% cellularity. Then, the cells were incubated at 37 °C with KaryoMAX^®^ Colcemid^TM^ (Life Technologies, Carlsbad, CA, USA) for four hours. After that, the sample was centrifuged, the supernatant was removed, and 6 mL of hypotonic solution (0.075 M KCl with HEPES) was added to the pellet and incubated for 20 min at 37 °C. The sample was centrifuged, and the cells were fixed with cold methanol-acetic acid solution (3:1 and 2:1—Merck, Darmstadt, Germany). Cleaned slides were placed in a water bath at 60 °C, in which we dripped the cell suspension. The slides were aged at 60 °C overnight. For G-banding, the slides were submitted to the G-banding method using trypsin (0.002 g/mL), washed in saline solution, and rinsed in distilled water. The slides were stained with Giemsa (1:20) solution (Laborclin, Paraná, Brazil) and evaluated under a Leica microscope (DM2000) (magnification: 100X). Twenty metaphases were analyzed, to detect numerical and/or structural aberrations. All the results were described according to the International System for Human Cytogenetic Nomenclature (2016) [78].

#### 4.3.6. Cytokinesis-Block Micronucleus Assay

The cytokinesis-block micronucleus (CBMN) assay was used to assess UC-MSC genotoxicity [70]. Cord blood mononuclear cells (CBMNCs) were used as a normal control, and HeLa cells were used as a positive control. Both cell types were used after thawing under the same conditions as the UC-MSCs. Twenty-four hours after cell expansion, confluence was evaluated, and 60% confluence was the ideal condition for this assay. Cytochalasin B (Sigma-Aldrich) was added to each T25 flask. After 24 h of incubation, the cells were enzymatically dissociated with 0.25% trypsin-EDTA, as described in Section 4.2. Hypotonic solution was added for 1 min, and the cells were fixed with cold methanol-acetic acid solution (9:1) (Merck, Darmstadt, Germany). For slide preparation, the supernatant was removed and centrifuged, and the cells were resuspended in 150 µL of methanol-acetic acid solution (9:1). The slides were placed in a water bath at 40 °C, and 30 µL of the cell suspension was distributed on them. The cells were aged in a 60 °C oven for 30 min and stained with a Giemsa solution (Laborclin). One thousand cells per sample were analyzed using 40X magnification on a Leica microscope (DM2000). The nuclear division index (NDI) was calculated after the analysis of each sample. The lowest NDI value possible is 1.0, which occurs when none of the cells had divided during the cytokinesis-block period; the highest NDI value is 2.0, considering that all of them are binucleated [83].

#### 4.3.7. Soft Agar Colony Formation Assay

Anchorage-independent growth was evaluated by a soft agar colony formation assay, which is a method for detecting the malignant transformation of cells. Soft agar colony formation was conducted using two methods: a standard soft agar assay and a CytoSelect™ 96-well cell transformation assay kit.

In the standard soft agar assay, the cells are allowed to grow inside semisolid culture media for three weeks [74]. This assay was performed on a six-well plate (TPP) precoated with a base agar layer composed of a 1% soft agar solution (Kasvi) in Dulbecco’s modified Eagle’s medium (DMEM) (Gibco) and 10% FBS. This underlayer was allowed to solidify before use. Then, 2 × 10^4^ UC-MSCs were suspended in DMEM containing FBS and 0.6% agar solution and plated onto precoated plates. HeLa cells were used as a positive control. All experiments were performed in technical duplicate. Plates were incubated at 37 °C with 5% CO_2_ for 21 days, after which colony formation was observed under a bright-field microscope.

Using the CytoSelect^TM^ 96-well cell transformation assay kit (Cell Biolabs, San Diego, CA, USA), a short incubation time is needed, and it is a quantitative method where cell growth is detected by a fluorescent dye [84]. This assay was performed following the manufacturer’s instructions. Briefly, as a base layer, 50 µL of 0.6% agar solution was transferred to each well of a 96-well flat-bottom microplate. The plate was transferred to 4 °C. for 30 min to allow the base agar layer to solidify. The cell agar top layer was prepared using equal volumes of 1.2% agar solution, 2X DMEM/20% FBS media, and concentrations of 1000, 5000, and 10,000 cell suspension—HeLa (positive control) or UC-MSCs. This mixture was transferred to each well of a 96-well plate. The cells were incubated for 7 days at 37 °C and 5% CO_2_. The agar was solubilized, and the cells were lysed. The DNA was stained with CyQuant GR dye, and the plate was read in a 96-well hybrid microplate reader using the following settings: 488 nm excitation, 530 nm emission, gain 100, and bottom optics (BioTek Synergy H1M2F, Agilent Technologies, Santa Clara, CA, USA).

#### 4.3.8. Potency Test—Inhibition of T-Lymphocyte Proliferation Assay

The immunomodulation potential was used as a potency test to evaluate the ability of UC-MSCs to inhibit the proliferation of T lymphocytes [85]. UC-MSCs were cocultured with peripheral blood mononuclear cells (PBMCs). Peripheral mononuclear cells were obtained from the peripheral blood of healthy donors and isolated using a Ficoll-Paque density gradient. Peripheral blood was diluted 1:3 with IMDM, carefully loaded onto Histopaque (1.077 g/mL) (Cytiva, Uppsala, Sweden), and centrifuged at 400× *g* for 30 min. The cells were washed twice with IMDM. A portion of PBMCs was labeled with 5 µM 5,6-carboxyfluorescein diacetate succinimidyl ester (CFSE) (Sigma–Aldrich, St. Louis, MO, EUA), and the other portion was used as a negative control. MSCs were plated (0.5 × 10^6^ cells/well) in triplicate in 24-well plates (TPP) and incubated for 6 h at 37 °C with 5% CO_2_. Labeled CFSE PBMCs were added to plates containing MSCs at ratios of 1:2, 1:5, and 1:10 (PBMC:MSC). Then, 0.5 µg/mL phytohemagglutinin (PHA) (Sigma–Aldrich) was added as a polyclonal stimulus for T lymphocyte proliferation. The plate was incubated at 37 °C for 5 days, and the cells were collected and marked for the detection of the CD3 APC marker (BD Pharmingen) by flow cytometry. The lymphocyte proliferation observed in the cocultures of CFSE-PBMCs plus MSCs was normalized to the control without MSCs, which was set as 100% proliferation. The percentage of inhibition of proliferation was obtained by subtracting T lymphocytes that proliferated in the presence of MSCs.

#### 4.3.9. Statistical Analysis

The Shapiro-Wilk test was used to verify the distribution of quantitative variables. Parametric data were analyzed using the unpaired t test (proliferation of T lymphocytes vs. control and soft agar colony formation assay), and nonparametric data were analyzed using the Mann-Whitney test (quantification of adipogenic and osteogenic differentiation, micronucleus assay). For multiple comparisons, the Kruskal–Wallis test (proliferation of T lymphocytes—different concentrations) was used. Values are presented as the mean ± standard deviation (SD). Statistical analysis was performed in GraphPad Prism version 9.0.0 for Windows software (GraphPad, La Jolla, CA, USA). A *p* < 0.05 was considered statistically significant.

## 5. Conclusions

This study shows the relevance of the manufacturing process and quality controls in reducing variability due to the heterogeneity between donors and reinforces the assessment of TF/CD142 expression as a criterion for intravascular MSC products. The exhibited data might be useful for the implementation and optimization of new analytical techniques and automated methods to improve safety by evaluating genomic stability and tumorigenicity, which are the major concerns related to MSC-based therapy.

## Figures and Tables

**Figure 1 ijms-24-12955-f001:**
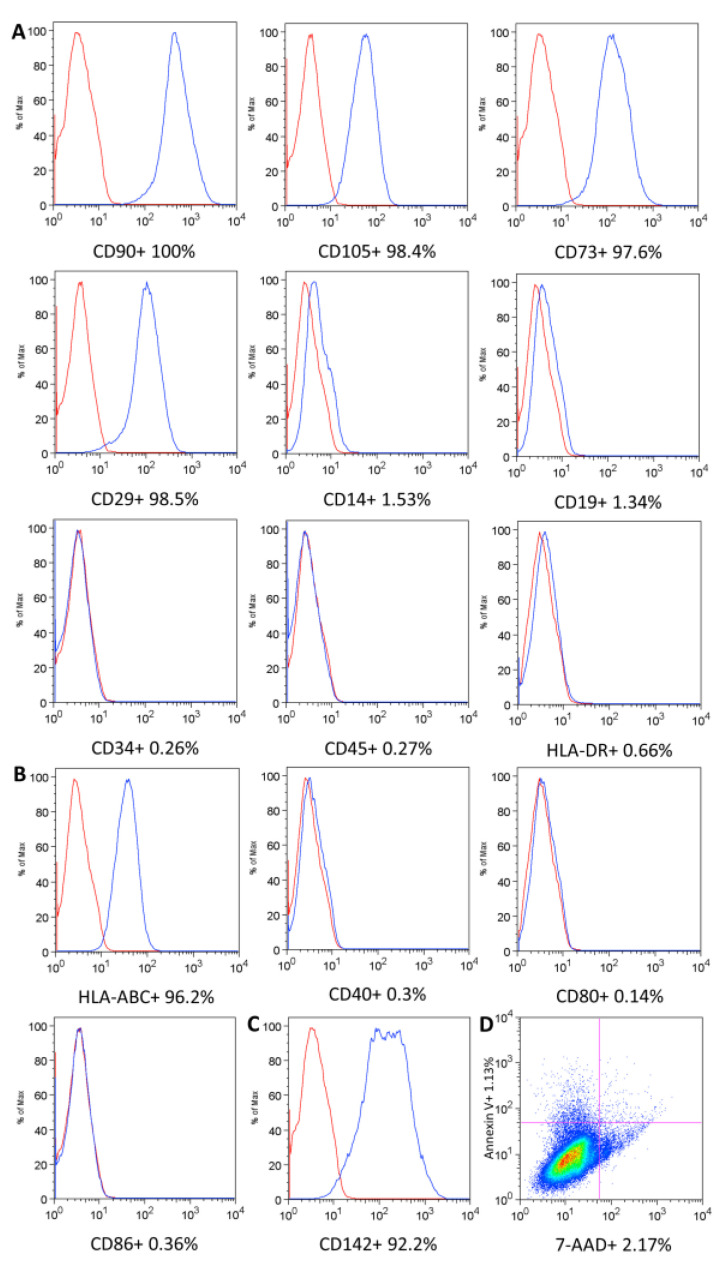
Umbilical cord MSC characterization. (**A**) UC-MSC surface markers. (**B**) Immunogenicity evaluation. (**C**) Procoagulant tissue factor. The isotype control is shown as a red line histogram. The surface markers are shown as a blue line histogram (**A**–**C**). Representative histograms. (**D**) Cell viability and apoptosis/necrosis.

**Figure 2 ijms-24-12955-f002:**
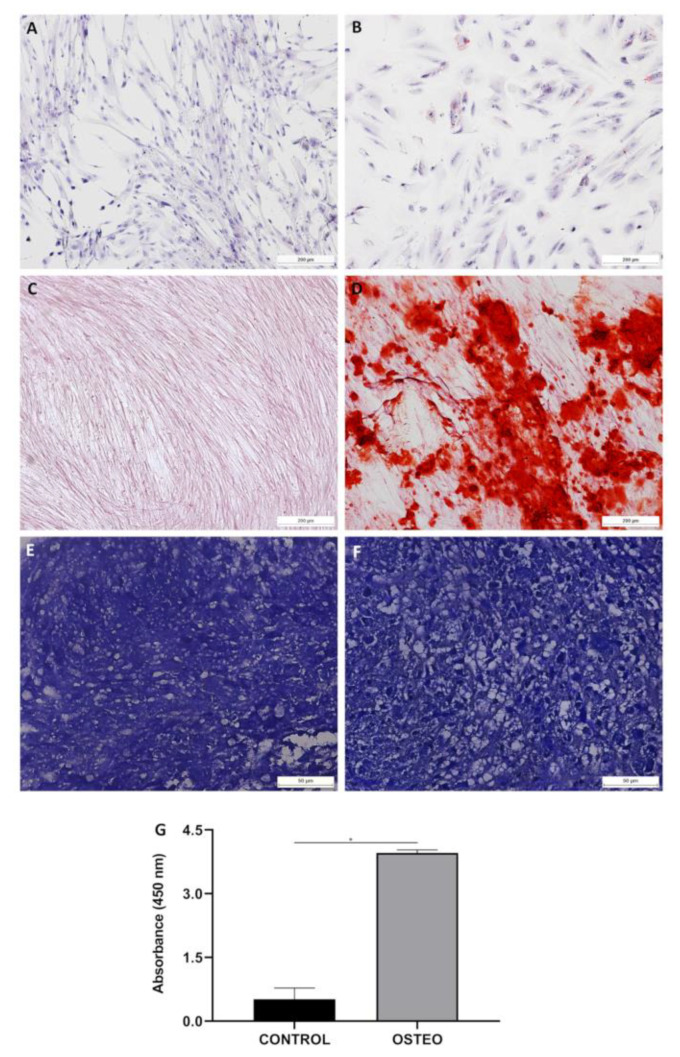
Mesenchymal stromal cell differentiation. (**A**,**C**,**E**) Control cells. (**B**) Cells differentiated into adipocytes, characterized by the presence of lipidic vacuoles stained with Oil Red O. (**D**) Cells differentiated into osteoblasts, characterized by the presence of calcium deposits stained with Alizarin Red S (red). (**F**) Cells differentiated into chondrocytes, characterized by the presence of lacunae around young chondrocytes and proteoglycan in the matrix. (**G**) Quantification of osteogenic differentiation. * Representative images. Error bars represent the SD of the measurements (*n* = 5). (**A**–**D**) Scale bar, 200 μm; (**E**,**F**) Scale bar, 50 μm.

**Figure 3 ijms-24-12955-f003:**
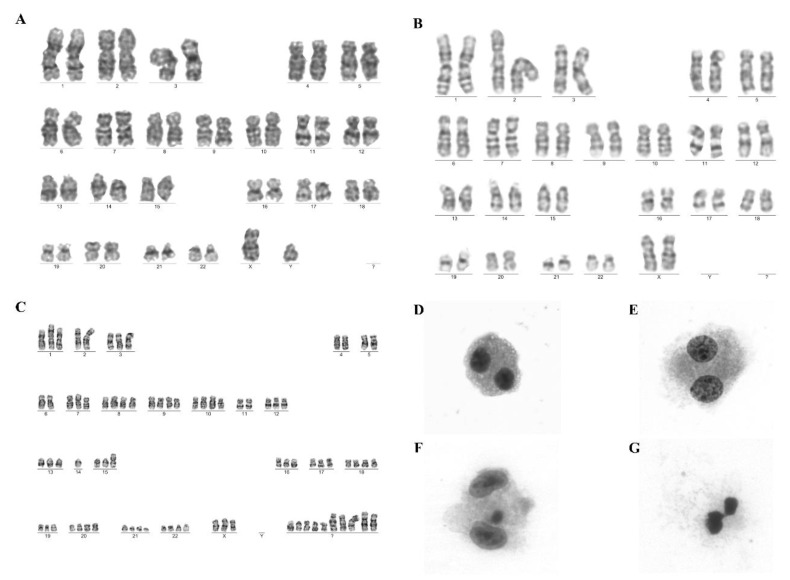
Genomic stability evaluation. (**A**–**C**) Representative images of UC-MSCs (**A**), CBMCs (**B**), and HeLa cells (**C**) showing a karyogram with chromosomal alterations analyzed by the G-banding technique. UC-MSC and CBMC showed normal complete karyograms (46, XY). (**D**–**G**) Binucleated cells analyzed by the CBMC assay (40X magnification). (**D**,**E**) CBMNCs showed normal binucleated cells. (**F**,**G**) HeLa cells showed binucleated cells with micronuclei (**F**) and nucleoplasmic bridges (**G**).

**Figure 4 ijms-24-12955-f004:**
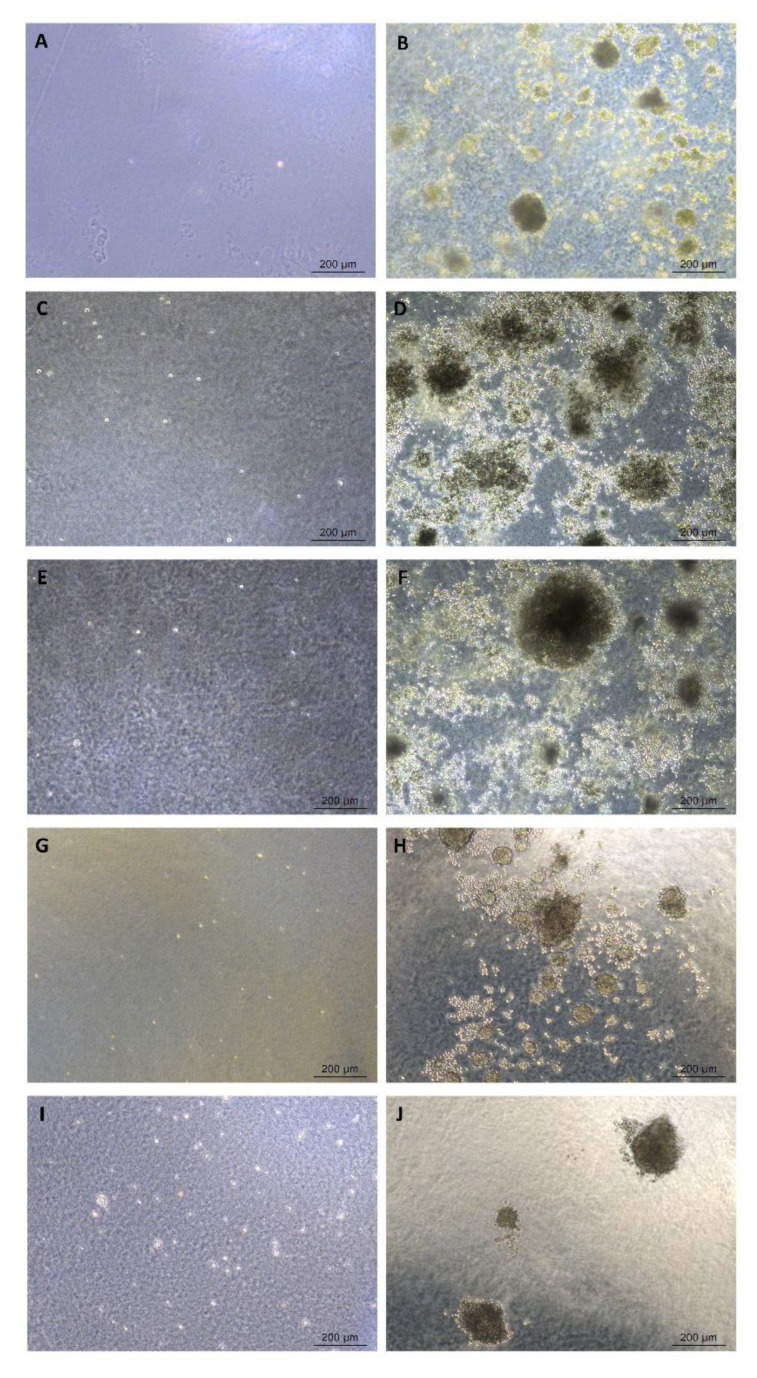
Soft agar transformation assay. Representative bright-field (BF) images of soft agar colony formation. UC-MSCs and HeLa cells were cultivated for 21 days on the agar. (**A**,**C**,**E**,**G**,**I**) Soft agar culture of UC-MSC samples did not generate any colonies (*n* = 5). (**B**,**D**,**F**,**H**,**J**) HeLa cells were used as positive control transformed cells, and they grew into colonies. Scale bar, 200 μm.

**Figure 5 ijms-24-12955-f005:**
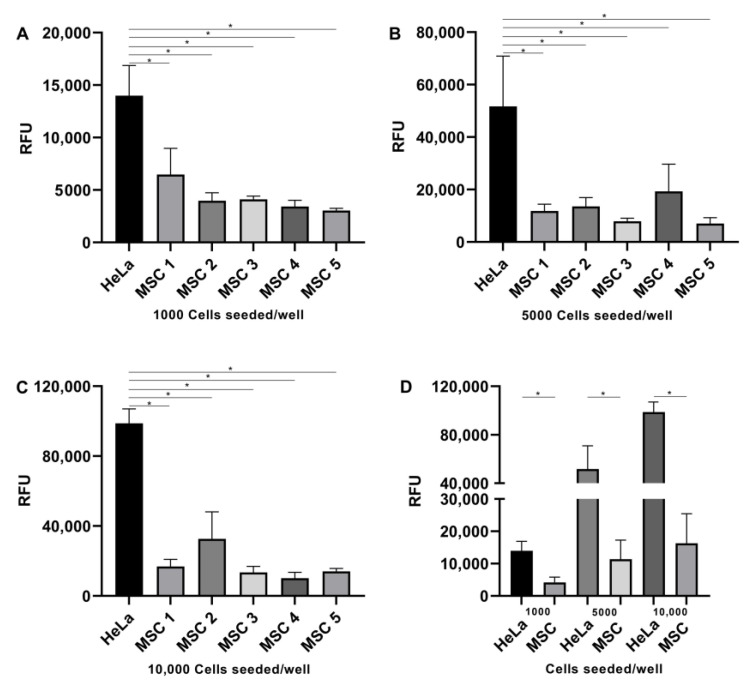
Quantitative assessment of colonies in the soft agar assay. (**A–D**) HeLa cells were used as a positive control, and five UC-MSC samples were plated at 1000, 5000, and 10,000 cells/well in soft agar media. After seven days of incubation, there was a significant difference in Relative Fluorescence Units (RFU) between HeLa cells and UC-MSCs. Error bars represent the SD of the measurements. * Statistical significance.

**Figure 6 ijms-24-12955-f006:**
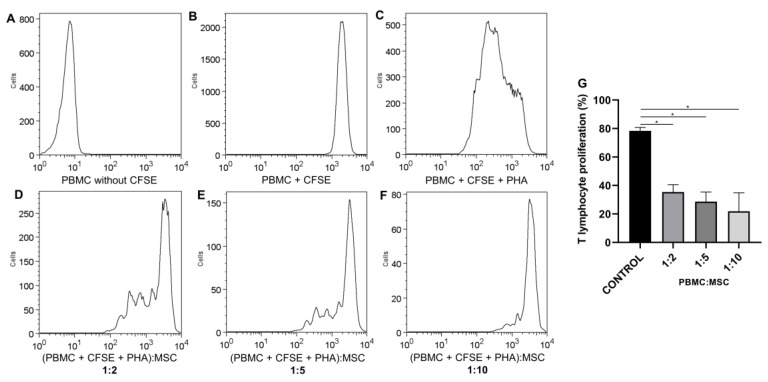
Inhibition of the T-lymphocyte proliferation assay; representative histograms. UC-MSCs were cultivated with PHA-stimulated CD3+ lymphocytes labeled with CFSE. (**A**) CD3+ lymphocytes not labeled with CFSE. (**B**) CFSE-labeled CD3+ lymphocytes. (**C**) CD3+ lymphocytes labeled with CFSE and stimulated with PHA. (**D**–**F**) CD3+ lymphocytes labeled with CFSE were cultivated with UC-MSCs at concentrations of 1:2, 1:5, and 1:10. (**G**) Percentage of lymphocyte proliferation at different concentrations compared to control cells (T lymphocyte + PHA). Error bars represent the SD of the measurements (*n* = 5). * Statistical significance.

**Figure 7 ijms-24-12955-f007:**
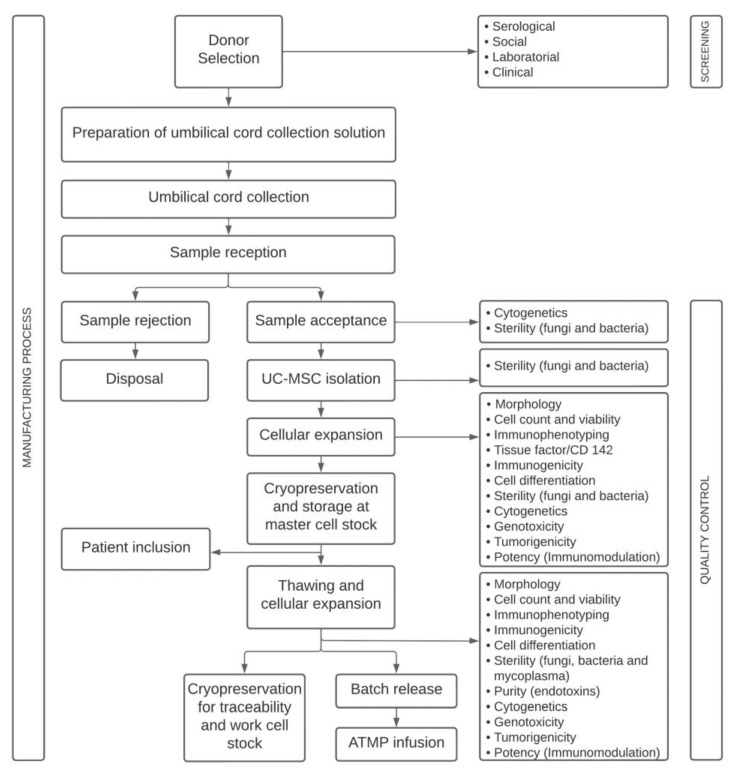
Flow diagram of the phases involved in the manufacturing of a UC-MSC-based product for clinical administration.

**Table 1 ijms-24-12955-t001:** Quality control carried out in different phases of UC-MSC manufacturing and acceptance criteria.

Process	Quality Controls	Methods	Acceptance Criteria	Reference
UC donor selection	Screening	N/A	Eligible donor	FDA Guidance for Industry—Eligibility Determination for Donors of Human Cells, Tissues, and Cellular and Tissue-Based Products
	Serology for HBV, HCV, HIV, HTLV, *Treponema pallidum* and *Trypanossoma cruzi*	Serology and nucleic acid detection	Negative	FDA Guidance for Industry—Eligibility Determination for Donors of Human Cells, Tissues, and Cellular and Tissue-Based Products RDC ANVISA (508/2021)—Good Manufacturing Practice (GMP) requirements for Therapeutic Use and Clinical Research [3]
UC isolation	Sterility	Automated growth-based	Absence of bacterial or fungal growth	Eur. Ph.: (2.6.27) Microbiological control of cellular products
UC-MSC expansion	Adherence to plastic	Microscope observation	Adherent	International Society for Cell & Gene Therapy (ISCT) [56]
	Viability	Flow cytometry	≥70%	FDA Guidance for Industry—Chemistry, Manufacturing, and Control (CMC) Information for Human Gene Therapy Investigational New Drug Applications
	Phenotypic markers	Flow cytometry	≥95% for CD90, CD105, CD73, CD29 and ≤2% for CD14, CD19, CD34, CD45, HLA-DR	International Society for Cell & Gene Therapy (ISCT) [56]
	Immunogenicity	Flow cytometry	≥90% for HLA-ABC ≤2% for CD40, CD80, CD86, HLA-DR	N/A
	Differential potential assay	Cell differentiation assay	Differentiate into osteoblasts, adipocytes and chondroblasts	International Society for Cell & Gene Therapy (ISCT) [56]
	Sterility	Automated growth-based	Absence of bacterial or fungal growth	Eur. Ph.: (2.6.27) Microbiological control of cellular products
	Genomic stability	G-Banding Technique	Absence clonal chromosomal alterations	International System for Human Cytogenomic Nomenclature (ISCN) [78]
	Potency	Inhibition of T-lymphocyte proliferation assay	Concentration 1:10 (PBMC:MSC) ≥ 50%	N/A
	Genotoxicity	Cytokinesis-block micronucleus assay	No significantly DNA damage events	N/A
	Tumorigenicity	Soft agar colony formation assay	Non-tumorigenic	N/A
UC-MSC final product	Adherence to plastic	Microscope observation	Adherent cells	International Society for Cell & Gene Therapy (ISCT) [56]
	Viability	Flow cytometry	≥70%	FDA Guidance for Industry—Chemistry, Manufacturing, and Control (CMC) Information for Human Gene Therapy Investigational New Drug Applications
	Phenotypic markers	Flow cytometry	≥95% for CD90, CD105, CD73, CD29 and ≤2% for CD14, CD19, CD34, CD45, HLA-DR	International Society for Cell & Gene Therapy (ISCT) [56]
	Procoagulant tissue factor (TF)	Flow cytometry	N/A	N/A
	Sterility	Automated growth-based	Absence of bacterial or fungal growth	Eur. Ph.: (2.6.27) Microbiological control of cellular products
	Mycoplasma	PCR-based and bioluminescent assays	Not detected	Eur. Ph. (2.6.7.) Monograph Mycoplasmas
	Purity	Chromogenic kinetic	<0.5 EU/mL	FDA Guidance for Industry—Chemistry, Manufacturing, and Control (CMC) Information for Human Gene Therapy Investigational New Drug Applications Eur. Ph. (5.1.10.) Guidelines for using the test for bacterial endotoxins
	Genomic stability	G-Banding Technique	Absence clonal chromosomal alterations	International System for Human Cytogenomic Nomenclature (ISCN) [78]

Abbreviations: Umbilical Cord-Mesenchymal Stromal Cell (UCT-MSC), Hepatitis B Virus (HBV), Hepatitis C Virus (HCV), Human Immunodeficiency Virus (HIV), Human T-Lymphotropic Vírus (HTLV), Tissue Factor (TF), Giemsa banding (G-banding), Polymerase Chain Reaction (PCR), *Limulus amebocyte* lysate (LAL), Cluster Of Differentiation (CD), Human Leukocyte Antigen (HLA), Peripheral Blood Mononuclear Cells (PBMC), EU (Endotoxin Units), Food and Drug Administration (FDA), Collegiate Board Resolution (RDC), Brazilian Health Regulatory Agency (ANVISA), Good Manufacturing Practice (GMP), Eur. Ph. (European Pharmacopoeia), International System for Human Cytogenomic Nomenclature (ISCN), Chemistry, Manufacturing, and Control (CMC), International Society for Cell & Gene Therapy (ISCT), Not applicable (N/A).

## Data Availability

The data generated or analyzed during this study are included in this article.

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
