# Peer review of "Quality Control Optimization for Minimizing Security Risks Associated with Mesenchymal Stromal Cell-Based Product Development"

_ijms, 2023, doi:10.3390/ijms241612955_

Round 1
Reviewer 1 Report
In this paper, the authors describe how to standardize and optimized methods to guarantee the reproducibility, safety, and efficacy of MSC-based products to be administered to patients in clinical practice.
What type of cytometry was used? Was a multiparametric gating operation performed? Or was each "CD" evaluated as a single positivity? Some antibodies appear to be associated with the same fluorochrome, so it is not clear to me how the MSC identification procedure was conducted.
Was the respective expression of CD4 and CD8 evaluated in the analysis of CD3-positive T lymphocytes? This is an important point as the two populations act at different times and in different ways.
Author Response
Dear reviewer,
we are grateful for the important suggestions in the manuscript. We made changes according to the suggestions to the main text and figures. Changes have been highlighted with tracked changes. If any alteration was not clear, please do not hesitate in contacting us with comments.
We did the English language edition and answered the questions.
Specific comments:
- What type of cytometry was used? Was a multiparametric gating operation performed? Or was each "CD" evaluated as a single positivity? Some antibodies appear to be associated with the same fluorochrome, so it is not clear to me how the MSC identification procedure was conducted. In the FSC-H x SSC-H graphs, shown in pseudocolor, a gating was performed on the population of interest, named R1. From R1, the surface marker expression was analyzed individually in the histogram to assess positivity. The markers used were conjugated with FITC, PE, PE-Cy5, or APC. In cytometry tubes, up to four simultaneous markers with different fluorochromes were used to avoid overlapping data.
2. Was the respective expression of CD4 and CD8 evaluated in the analysis of CD3-positive T lymphocytes? This is an important point as the two populations act at different times and in different ways. CD4+ and CD8+ expression was not determined in the CD3+ T lymphocyte analysis of the flow cytometry potency test. To carry out this test, there was no need to identify the lymphocyte subpopulations after thawing; MSCs samples were in standard cell culture conditions and were not subjected to any stimulus or treatment that could modify the expression of CD4+ and CD8+.
Reviewer 2 Report
With this manuscript the authors present a protocol for quality control of cultured hMSCs originating from umbilical cord (UC-MSCs). The protocol is presented in form of exemplar results obtained from five bio-bank samples (i.e. excluding the methods of sample preparation, cell extraction aso.).
The authors conclude that this protocol shows the need for standardization, as differences between individual donors were found. The authors also emphasize the importance of the utilization of TF/CD142 as a marker for hMSC products.
The English text is quite good, a minor editorial check may be sufficient.
This study represents a real challenge for a reviewer, because there are multiple layers of relevance connected to the manuscript and for each of them the conclusions of the review may be interpreted as either approval or rejection accordingly. Therefore, to avoid inappropriate conclusions and interpretations, this review aims to treat each layer separately, as concise as possible.
First in regard of the solely scientific value of this study, we have to look at the new and innovative points introduced and how the study represents a valuable addition to existing knowledge. Here, in my opinion, the study falls short, as there is already numerous literature available on this topic of which some, but not all, highly relevant papers have been cited by the authors (http://dx.doi.org/10.1016/j.jcyt.2016.05.012, https://doi.org/10.3390/cells11172732, https://doi.org/101186/s13287-020-01696-6). No new technology or methods are introduced. Also the utilization of TF/CD142 for characterization of hMSCs has already been strongly recommended in this context (https://doi.org/10.1016/j.molmed.2018.12.006 and https://doi.org/10.1093/stcltm/szab005). The authors are highly encouraged to include in their response their view about the additional scientific value of this study, in case their perspective on purely scientific relevance is different.
The second layer is regarding the experimental quality and the general suitability of applied methods for characterization of hMSCs. The experimental quality appears overall to be quite good. There are some questions regarding analysis:
Average cell viability is described in the abstract with 70% and in line 102 with 96.9%. This may be also be misleading, as the viability is very likely the viability after thawing or even in common cell culture and not after preparation. This should be made clear in the text.
The annexin V quantification is described as "value" (line 103), but no unit or indication of the method of measurement (i.e. adsorption/arbitrary values) is provided.
For visualizing cell viability a cytometric 2D-plot showing both the population clouds of viable and non-viable cells should be provided, to show how well separated the two populations are. This is actually required/desired for all cytometric markers.
Osteogenic quantification is either not well described and/or significantly deviant to common protocols. First, there is no description of pH-adjustments (acetic acid, ammonium hydroxide) in the material and method section and the absorbance (see: https://doi.org/10.3390/ijms24010723) at 445 appears to be suboptimal, according to literature.
Generally, the material and methods section is not very detailed in regards of antibodies used, fluorescence markers and material sources. For example, when reporting "enzymatic dissociation" the enzyme used its concentration and source as well as other related conditions should be provided.
The next level regards the validity of the proposed protocol (without consideration of potential requirements of regulatory bodies). Here, it appears to be clear that this study is starting, according to Figure 1, after thawing and cellular expansion (line 96). So it is only treating a single section of the whole manufacturing process, a step that is however very critical and important. Nevertheless, depending on the projected level of security, quality control steps are also recommended at other stages of manufacture, as is clearly demonstrated when comparing Figure 1 of this manuscript to Figure 2 in doi:10.3390/pharmaceutics11110552. Another difference observed is, that many existing protocols are designed to test for a specific set of characteristics that is then provided as product description/ data sheet. There are some distributed mentions of product requirements in this manuscript, but I miss a clear product description, at best in form of a table. One example for this description may be found in table 1 of http://dx.doi.org/10.1016/j.jcyt.2016.05.012. If provided, clinicians using these cells can then decide, if the provided set of characteristics that is guaranteed for by quality control, is fitting to their specific requirements or not.
The next level regards the question, if the provided protocol is in line with the requirement of regulatory bodies regarding product safety. The task of scientific reviewer cannot be to declare the given protocol as "save for application" or in "accordance with regulatory requirements". One additional reason is that the protocol is finally not complete, as pointed out above, but also because of insufficient knowledge about actual requirements in Brazil or Parana on the side of the reviewer.
The authors mention "Manufacturing Process Standardization" in their title, but what actually happens is that many laboratories in the world come up with their own manufacturing protocol, leading to a increasing diversification of standards. In my personal opinion, the way of very clear and rich product definition/description/data sheets linked to according protocols and optimized safety steps at all production levels is the way to go. So it would not end in one optimal protocol, but one protocol for only one very specific product.
The reviewer is not author, co-author or otherwise related to any of the references provided in this review.
The English language is good, no severe problems in grammar, language or orthography were detected. However, a minor editorial checkup is recommeded, as the reviewer is also not a native English speaker.
Author Response
Dear reviewer,
we are grateful for the important suggestions in the manuscript. We made changes according to the suggestions to the abstract, main text, figures and tables. Changes have been highlighted with tracked changes. If any alteration was not clear, please do not hesitate in contacting us with comments.
We did the English language edition and answered the questions.
Specific comments:
- The authors are highly encouraged to include in their response their view about the additional scientific value of this study, in case their perspective on purely scientific relevance is different.
We agree with this observation; the scientific contribution of this study in the area of Cell Therapy was unclear. New paragraphs have been added in the introduction (line 93) and discussion (lines 250, 257, 337 and 411).
- Average cell viability is described in the abstract with 70% and in line 102 with 96.9%. This may be also be misleading, as the viability is very likely the viability after thawing or even in common cell culture and not after preparation. This should be made clear in the text.
Viability was evaluated in samples in cell culture approximately seven days after thawing. The average results obtained was 96.9% ± 1,715, as indicated in line 103. In the abstract, we incorrectly placed the viability data. We appreciate the reviewer's note and have corrected this information as noted above.
According to the “Chemistry, Manufacturing, and Control (CMC) Information for Human Gene Therapy Investigational New Drug Applications (INDs)” available in the “Cellular & Gene Therapy Guidances” of the U.S. Food and Drug Administration (FDA) is recommended as a minimum acceptable viability of at least 70% for modified cells. This reference value can be applied to MSCs, as it is understood that dead cells and cell debris in the sample do not affect the safe infusion of the product in patients and/or the therapeutic effect.
Reference - U.S. Food and Drug Administration (FDA) - https://www.fda.gov/vaccines-blood-biologics/biologics-guidances/cellular-gene-therapy-guidances
- The annexin V quantification is described as "value" (line 103), but no unit or indication of the method of measurement (i.e. adsorption/arbitrary values) is provided.
Annexin V quantification was determined by combined analysis with the vital dye 7-AAD. Cells in apoptosis were considered positive for Annexin V and negative for 7-AAD. The results obtained were expressed in percentages. Pseudocolor plot graph was included with cell viability results. A new figure has been added to the manuscript (Figure 2).
- For visualizing cell viability a cytometric 2D-plot showing both the population clouds of viable and non-viable cells should be provided, to show how well separated the two populations are. This is actually required/desired for all cytometric markers.
As requested by the reviewer, we modified the figures and replaced the graphs with a cytometric 2D plot to demonstrate the expression of the markers of the UC-MSC, immunogenicity e o procoagulant tissue factor (TF). The pseudocolor plot graph was also included with cell viability results. A new figure has been added to the manuscript (Figure 2).
- Osteogenic quantification is either not well described and/or significantly deviant to common protocols. First, there is no description of pH-adjustments (acetic acid, ammonium hydroxide) in the material and method section and the absorbance (see: https://doi.org/10.3390/ijms24010723) at 445 appears to be suboptimal, according to literature.
The osteogenic quantification protocol was more detailed in material and methods, as requested by the reviewer. The alizarin red assay is the most used to quantify osteogenic differentiation, and there may be a need to adapt the protocol according to the evaluated samples. Umbilical cord MSC is easily differentiated into osteoblasts. Therefore, it was possible to quantify the calcium crystals in the samples effectively. We observed a different method in the article suggested by the reviewer, which aimed to increase the mineralization capacity of human and murine osteoblasts to optimize the signal in the alizarin red assay. The techniques are different (procedure, incubations, and some reagents), but this study and ours allowed the analysis of osteogenic lineage.
- Generally, the material and methods section is not very detailed in regards of antibodies used, fluorescence markers and material sources. For example, when reporting "enzymatic dissociation" the enzyme used its concentration and source as well as other related conditions should be provided.
The material and methods section was improved.
- The next level regards the validity of the proposed protocol (without consideration of potential requirements of regulatory bodies). Here, it appears to be clear that this study is starting, according to Figure 1, after thawing and cellular expansion (line 96). So it is only treating a single section of the whole manufacturing process, a step that is however very critical and important.
We appreciate the suggestion. Figure 1 was remade, and all phases of the manufacturing process and quality controls are shown.
- Another difference observed is, that many existing protocols are designed to test for a specific set of characteristics that is then provided as product description/ data sheet. There are some distributed mentions of product requirements in this manuscript, but I miss a clear product description, at best in form of a table.
We agree with this observation. A table with process, quality controls, methods, release criteria, reference, and results for each batch was added.
- The next level regards the question, if the provided protocol is in line with the requirement of regulatory bodies regarding product safety. The task of scientific reviewer cannot be to declare the given protocol as "save for application" or in "accordance with regulatory requirements". One additional reason is that the protocol is finally not complete, as pointed out above, but also because of insufficient knowledge about actual requirements in Brazil or Parana on the side of the reviewer.
Quality controls to ensure product safety are described in the rules of Brazilian regulation for ATMPs, according to reference [3]. In this study, all the quality controls required by Brazilian legislation were performed, and additional tests were proposed, such as the evaluation of procoagulant tissue factor (TF), genotoxicity, and tumorigenicity to reduce the risks associated with the infusion of these cells.
According to Brazilian legislation, ATMP can only be released for Therapeutic Use and clinical research after performing the following tests: a) count of the total number of relevant cells; b) appropriate identity or phenotyping test for the product and quantification of the cell populations present; c) cell viability; d) purity test: verification of the presence of endotoxins; e) microbiological tests to detect bacterial contamination (aerobic and anaerobic) and fungal, test for detection of mycoplasma contamination; g) cytogenetics; h) potency tests.
Reference 3. ANVISA Brazilian Health Regulatory Agency. Resolution RDC No 508, 27 May 2021. Provides for good practices in human cells for therapeutic use and clinical research and makes other provisions. Available online: https://bvsms.saude.gov.br/bvs/saudelegis/anvisa/2020/rdc0508_27_05_2021.pdf (accessed on 01 February 2023).
- The authors mention "Manufacturing Process Standardization" in their title, but what actually happens is that many laboratories in the world come up with their own manufacturing protocol, leading to an increasing diversification of standards. In my personal opinion, the way of very clear and rich product definition/description/data sheets linked to according protocols and optimized safety steps at all production levels is the way to go. So it would not end in one optimal protocol, but one protocol for only one very specific product.
We agree with this observation and modified the manuscript's title to “Quality Control Optimization to Minimize Security Risks Associated with Mesenchymal Stromal Cell-Based Product Development”.
Round 2
Reviewer 1 Report
The authors better define the experimental part of cytometry as regards the gating procedure and the use of fluorochromes for the identification of the cell population of interest
Author Response
We are grateful for the important suggestions of reviewer 1. We made changes in the cytometry methodology. Therefore, we do not understand the reviewer's comments in round 2.
In round 1, the reviewer suggested that the introduction, research design, and methods could be improved. All these sessions have been improved, and we respond to the comments. However, in round 2, the reviewer said we answered the comments but stated that all criteria, such as introduction, references, research design, methods, results, and conclusions, must be improved. Could there have been a mistake? It would help us if the reviewer could highlight the points that should be improved.
Reviewer 2 Report
As already pointed out in my first review, this manuscript has a special scope. It does not investigate a testable hypothesis, but presents a protocol for producing a cell therapy product that is intended to be used for human medical treatment. The quality of the presented protocol appears to be at state-of-the-art and the authors did include quality control steps at several points of production in the revised version. The documentation of production and testing is overall sound and no critical flaws have been detected. From an experimental view, the manuscript can be recommended for publication. I would nevertheless like to emphasize, that this approval of the protocol is only regarding its suitability for publication.
The proposed protocol appears to be well designed and the innovative/scientific added value is stated to be represented by an improvement of cell therapy product quality. However, while it is possible to review the experimental quality within the presented manuscript, a comprehensive comparative analysis including competitive protocols is beyond the scope of this review and it can therefore not be concluded, that this protocol introduces a generally improved approach. It has finally to be an editorial decision, if such a "protocol report" is suitable for publication in IJMS.
The authors did respond to all points of criticism raised in the first review of the manuscript.
The English language in the manuscript is overall good, but I would recommend an additional checkup of language and style by a native speaker on editorial level before publication.
Author Response
We are grateful for the valuable comments on our manuscript. It has improved a lot with the changes we have made.
As mentioned in our comments to the editor, we would like to emphasize the scientific and innovative value of this study.
Quality control of mesenchymal stem cell-based product development is crucially needed to deliver well-characterized and safe cell therapy products for clinical use. The regulations referring to Advanced Therapy Medicinal Products (ATMP) define the need to perform quality controls to guarantee the product´s safety. However, it does not define in which steps they must be performed and the methodologies that must be used to evaluate the different parameters.
This manuscript describes the cell therapy products pipeline. It makes essential contributions, showing that the protocol used (set of techniques) for quality control of ATMP in different steps of the manufacturing process allows reproducibility and meets all requirements of safety criteria established by Brazilian and international regulations. New quality control methodologies were also supplied to reduce the risks associated with product manufacture.
There are few scientific reports on the application of regulatory criteria in ATMP quality control. Therefore, this article addresses the translational gap between regulatory frameworks related to ATMP and laboratory practice, bringing important contributions to the clinical development of ATMP.
The manufacturing protocol used in this study made it possible to obtain a final product that meets the Brazilian regulation criteria and that can be replicated in other scenarios and cell therapy products for the treatment of inflammatory diseases and pneumonia.
Our aim was to share information to allow the practice of open science and contribute to advances in the area of cell therapy.
To clarify this to the reader, a paragraph was included in the discussion session:
“This manuscript describes the cell therapy products pipeline. It makes essential contributions, showing that the protocol used for quality control of ATMP in different steps of the manufacturing process allows reproducibility and meets all requirements of safety criteria established by Brazilian and international regulations.
There are few scientific reports on the application of regulatory criteria in ATMP quality control. Therefore, this research addresses the translational gap between regulatory frameworks related to ATMP and laboratory practice, bringing important contributions to the clinical development of ATMP.”
We performed English language editing, and changes have been highlighted with tracked changes.
Round 3
Reviewer 1 Report
no comments